



# Evaluation of MOPITT and TROPOMI carbon monoxide retrievals using AirCore *in situ* vertical profiles

Sara Martínez-Alonso[1], Ilse Aben[2], Bianca C. Baier[3,4], Tobias Borsdorff[2], Merritt N. Deeter[1], Kathryn McKain[3,4], Colm Sweeney[3], and Helen Worden[1]

[1]Atmospheric Chemistry Observations and Modeling (ACOM), National Center for Atmospheric Research(NCAR), Boulder, CO, USA
[2]SRON Netherlands Institute for Space Research, Leiden, Netherlands
[3]Global Monitoring Laboratory (GML), National Oceanic and Atmospheric Administration, Boulder, CO, USA
[4]Cooperative Institute for Research in Environmental Sciences (CIRES), University of Colorado, Boulder, CO, USA

**Correspondence:** Sara Martínez-Alonso (sma@ucar.edu)

**Abstract.** AirCore *in situ* vertical profiles sample the atmosphere from near the surface to the lower stratosphere, making them ideal for the validation of satellite tropospheric trace gas data. Here we present intercomparison results of AirCore carbon monoxide (CO) measurements with respect to retrievals from MOPITT (Measurements of Pollution In The Troposphere; version 8) and TROPOMI (TROPOspheric Monitoring Instrument), onboard the NASA Terra and ESA Sentinel 5-Precursor satellites, respectively. Mean MOPITT/AirCore total column bias values and their standard deviation ($0.0\pm0.9$, $0.3\pm1.0$, and $0.1\pm1.0$ for MOPITT thermal-infrared, near-infrared, and multispectral retrievals, respectively; all in units of $10^{17}$ molec cm$^{-2}$) are similar to results obtained in MOPITT/NOAA aircraft flask data comparisons from this study and from previous validation efforts. MOPITT CO retrievals are systematically validated using *in situ* vertical profiles from a variety of aircraft campaigns. Because most aircraft vertical profiles do not sample the troposphere's entire vertical extent, they must be extended upwards in order to be usable in validation. Here we quantify for the first time the error introduced in MOPITT CO validation by the use of shorter aircraft vertical profiles extended upwards by analyzing validation results from AirCore CO vertical profiles. Our results indicate that the error is small, affects mostly upper tropospheric retrievals (at 300 hPa: ~2.6, 0.8, and 3.2 percent points for MOPITT thermal-infrared, near-infrared, and multispectral, respectively), and may have resulted in the overestimation of MOPITT retrieval biases in that region. TROPOMI can retrieve CO under both clear and cloudy conditions. The latter is achieved by quantifying interfering trace gases and parameters describing the cloud contamination of the measurements together with the CO column; then, the reference CO profiles used in the retrieval are scaled based on estimated above-cloud CO rather than on estimated total CO. We use AirCore measurements to evaluate the error introduced by this approach in cloudy TROPOMI retrievals over land after accounting for TROPOMI's vertical sensitivity to CO (relative bias and its standard deviation = 2.02 % ± 11.13 %). We also quantify the null-space error, which accounts for differences between the shape of TROPOMI reference profiles and that of AirCore true profiles (for TROPOMI cloudy $e_{null}$ = 0.98 % ± 2.32 %).



# 1 Introduction

Tropospheric CO (carbon monoxide) is mostly produced by incomplete combustion of fuels, biomass burning, and oxidation of $CH_4$ (methane) and other hydrocarbons. Its main sink is oxidation by OH (the hydroxyl radical) (Spivakovsky et al., 2000; Lelieveld et al., 2016). CO is of great importance for the understanding of climate and for monitoring and predicting air quality because it has an indirect positive radiative forcing and is an excellent tracer to identify pollution sources, transport, and sinks. A long, consistent global tropospheric CO record allows for the detection of spatial, seasonal, and long-term trends as well as for the placement of individual CO-emitting events into context, key to a better understanding of their significance.

The MOPITT (Measurements of Pollution In The Troposphere) instrument, onboard NASA's Terra satellite, provides the longest global record of tropospheric CO available to date (2000-present). The MOPITT dataset is consistent and, thus, useful in climate and air quality analyses because it is systematically validated with respect to both aircraft data (Emmons et al., 2004; Deeter et al., 2010, 2012, 2013, 2014, 2017, 2018, 2019) and ground-based measurements (Buchholz et al., 2017; Hedelius et al., 2019). The vertical extent of most aircraft *in situ* profiles used to validate satellite retrievals of tropospheric trace gases is largely determined by the range of the aircraft used to collect them, and is often not sufficient to sample in its entirety the vertical column sensed by the satellite instruments. In those cases the aircraft measurements closest to the tropopause and modeled *a priori* vertical profiles of the species of interest are used to extend upwards the measured aircraft profiles to allow for comparison to satellite retrieved values; the error associated with this approach is unknown.

TROPOMI (the TROPOspheric Monitoring Instrument), onboard the ESA Sentinel-5 Precursor platform, measures CO, among other species, at high spatial resolution. Unlike MOPITT, TROPOMI can retrieve CO under both clear and cloudy conditions. TROPOMI retrieves parameters describing the cloud contamination of the measurement (cloud height, cloud optical thickness) and interfering trace gases together with the CO column. TROPOMI retrievals are based on the profile scaling method (Borsdorff et al., 2014; Landgraf et al., 2016). Under cloudy conditions the scaling of the profile is estimated by the CO concentration in higher altitudes in the atmosphere instead of the real total CO column; this is fully described by the total column AK (averaging kernel) supplied with the data product. The error introduced by this approach on cloudy TROPOMI retrievals over bodies of water has been previously quantified (on the order of a few percent; Martínez-Alonso et al., 2020). Errors over land could in theory be larger, since most pollution sources are on land and close to the surface (i.e., below cloud top).

Here we use AirCore data to estimate for the first time the error introduced in MOPITT validation results by the use of shorter aircraft vertical CO profiles extended upwards. We also investigate the error introduced by clouds on TROPOMI land CO retrievals by comparing them to AirCore vertical profiles. AirCore provides calibrated, high-precision measurements of CO and other long-lived species along vertical profiles from near the surface to the lower stratosphere. Because of its ability to sample such a large vertical range, AirCore is a great candidate for validating tropospheric satellite instruments. The AirCore atmospheric sampling system consists of an airborne coiled tube, typically flown on a balloon, filled with a gas of known composition which is evacuated during ascent; once the balloon altitude ceiling is reached, the now empty tube starts





a parachute-based descent during which it fills with the air it encounters. After recovery, the whole-air sample collected is analyzed in the laboratory for various long-lived atmospheric trace gases.

In the following sections we describe the datasets used in this study (Sect. 2), detail the methodology used in the analyses outlined above (Sect. 3), present our results (Sect. 4), and discuss their relevance (Sect. 5). In Section 6 we offer conclusions.

## 2 Datasets

### 2.1 AirCore

AirCore (Tans, 2009; Karion et al., 2010) is an innovative atmospheric sampling system that uses a long (~100 m), small-diameter (0.32 cm) coil of thin-walled stainless-steel tubing carried aloft on high-altitude balloons. Before launch the AirCore is filled with a gas mixture of known composition: the "fill gas", comprised of ambient levels of $CO_2$ (carbon dioxide) and $CH_4$, but spiked with high CO mole fraction. With one end closed and the other open to the outside air, the AirCore evacuates
the fill gas as the balloon ascends to ~30 km above mean sea level. Once the AirCore is released from the balloon, it collects a continuous sample of ambient air as it descends from the altitude ceiling to the ground. Upon landing, the open end of the coil is automatically closed, thus preserving the sample air inside. Mixing (which is only a result of molecular diffusion and Taylor dispersion) is relatively insignificant, ~0.5 m in both directions over the 4-hour typically needed to retrieve and analyze the air sample; thus, approximately 100 discrete samples can be measured in a coil of tubing 100 m long (Tans, 2021). The altitude
uncertainty of trace gas profiles retrieved using this technique, provided in the data files, is dependent upon the bi-directional diffusion of molecules of a gas of interest in the AirCore sample, and is larger at higher altitudes because the air sampled first (i.e., that in the stratosphere) has a longer diffusion time in the tubing coil. AirCore sample trace gas profiles retrieved are calibrated and traceable to World Meteorological Organization standard scales. The AirCore trace gas measurements have been rigorously evaluated and have shown comparable repeatability (precision) to those from aircraft data collected from continuous
analyzers and sampled in silicate glass flasks (Karion et al., 2010). The most recent AirCore dataset (13 August 2021 version; Baier et al., 2021) contains over 130 vertical profiles of CO, $CO_2$, $CH_4$, temperature, and relative humidity acquired at several locations worldwide (Fig. 1) between January 2012 and July 2021. Unlike most aircraft vertical profiles, AirCore profiles sample from near the surface to the lower stratosphere and, therefore, do not need to be extended upwards with their closest measurement to the tropopause and *a priori* values in order to represent the full tropospheric column as measured by satellite
instruments.

Current techniques for retrieving trace gas profiles rely on the use of a fill gas spiked, as explained earlier, with high CO mole fractions – all of which except ~1 % evacuates during balloon ascent – and a high CO mole fraction "push gas" that follows the AirCore sample during analysis. Both mixtures are used to identify the beginning and end of the air sample collected, but affect the topmost (stratospheric) and bottommost (near-surface) portion of the profile through end-member
mixing. With this method, CO is used to correct for "end-member" mixing in other trace gas profiles (Karion et al., 2010). The top- and bottommost portions of AirCore CO profiles used for this correction are thus discarded, resulting in CO profiles that extend from the near-surface to ~18-20 km above mean sea level. While CO in AirCore samples is measured by cavity





ringdown-spectroscopy (CRDS) and the measurement precision varies with CRDS analyzer, the total uncertainty is typically <5 ppb (Karion et al., 2013); AirCore CO is, however, still considered a developmental product. In addition, comparisons of
stratospheric CO profiles have shown differences up to ~15 ppb (Chen et al., 2022, in prep.), which could be a result of AirCore surface effects, chemical interactions or measurement interferences from other trace gas species, or incorrect AirCore sample end-member assumptions made.

The NOAA AirCore systems are typically deployed in pairs on the same balloon flight string. We have quantified the repeatability of retrieved CO profiles by comparing the 41 pairs of AirCore profiles launched simultaneously (i.e., with zero
minutes lag time) and from the same site. Each profile was resampled to a common 20,000-level vertical grid and intra-pair differences were calculated. Figure 2 shows that, at most altitudes, mean differences are well below ± 2 ppb (average 0.03 ppb). Mean differences at the top of the profiles (between 50 and 70 hPa, approximately) are slightly larger (5-15 ppb); this is consistent with the higher uncertainty in AirCore stratospheric CO retrievals described by Chen et al. (2022, in prep.).

We have compared the AirCore retrieved CO profiles to colocated CO vertical profiles from the NOAA aircraft flask dataset
(GLOBALVIEWplus v2.0 ObsPack; Sweeney et al., 2021) to quantify biases of the former with respect to the later (~accuracy). The NOAA aircraft flask dataset (or "aircraft dataset", for simplicity) has been described in detail by Sweeney et al. (2015) and used extensively in MOPITT validation (Deeter et al., 2019, and references therein). We analyzed the two datasets in their entirety (i.e., all sites and dates). Only profiles from the Southern Great Plains site (in Oklahoma, USA; 36.607° N, -97.489° E) acquired between January 2012 and July 2018 satisfied the different colocation criteria imposed. The averaged biases (AirCore
minus aircraft data) for the 5 available colocated pairs acquired less than 2 hours and 15 km apart range approximately between -6 and +6 ppb (near 750 and 920 hPa, respectively), with a 0.6 ppb overall average bias (Fig. 3.a). Allowing larger distances between colocated pairs results in more colocated pairs and a slight increase in biases closer to the surface: up to to 13 ppb for colocation distance <25 km (Fig. 3.b) and up to 20 ppb for colocation distance <50 km (Fig. 3.c) at ~875 hPa in both cases. Biases at lower pressure levels remain similarly low to those obtained with the most restricted colocation thresholds.
Increasing biases near the surface with larger colocation distances is consistent with CO values being more variable near the surface, where emissions take place. Horizontal displacements between the start and end of AirCore profiles are on average 26 km ± 10 km, similar to those between the start and end of NOAA aircraft flask profiles (14 km ± 18 km at the Park Falls, Wisconsin site; 53 km ± 28 km at the East Trout Lake, Saskatchewan site).

## 2.2 MOPITT

MOPITT, onboard NASA's Terra satellite, is a cross-track scanning gas correlation radiometer (Drummond and Mand, 1996; Drummond et al., 2010; Worden et al., 2013). From its sun-synchronous orbit at 705 km of altitude and 10:30 LST (local standard time) Equator crossing time, it provides global coverage approximately every 3 days with a 22 x 22 $km^2$ footprint at nadir. It measures radiances in two spectral bands: one in the near infrared (NIR, at ~2.3 $\mu$m), the other in the thermal infrared (TIR, at ~4.7 $\mu$m). Tropospheric CO profiles and total CO column values are derived separately from measurements
in each of these two bands, as well as from their combined multispectral radiances (TIR+NIR). MOPITT is currently the only satellite instrument capable of multispectral CO retrievals, which have enhanced sensitivity to CO near the surface in some



land observations (Worden et al., 2010). MOPITT CO profiles are provided for 10 levels (surface, 900 hPa, ..., 100 hPa) where each retrieval level corresponds to a uniformly weighted layer immediately above that level (Deeter et al., 2013). MOPITT retrievals are performed under clear conditions only, allowing ≤ 5 % cloud areal coverage inside the field-of-view. Here we use

level 2 TIR, NIR, and multispectral standard archival files (Deeter et al., 2017) from MOPITT version 8 (Deeter et al., 2019).

## 2.3 TROPOMI

TROPOMI is a push-broom imaging spectrometer in a sun-synchronous orbit at 824 km of altitude and with a 13:30 LST Equator-crossing time (Veefkind et al., 2012). Because of its wide (2600 km) swath width, it provides quasi-global daily coverage. Its spatial resolution at nadir is near 7 x 5.5 km$^2$ (across x along track) since 6 August 2019, down from around 7

x 7 km$^2$ before that date. A change in the Copernicus Sentinel-5P operations scenario resulted in this resolution improvement (Landgraf et al., 2021). TROPOMI measures radiance in the ultraviolet, visible, and reflected infrared; total CO column values are retrieved from the latter (from a ~2.3 $\mu$m band, like MOPITT). CO retrievals over land are obtained in both clear and cloudy conditions; the latter is possible by retrieving effective parameters (cloud height and optical thickness) that describe the cloud contamination of the measurements simultaneously with the trace gas columns (Landgraf et al., 2016) and then approximating

partial CO columns under cloud tops with scaled reference profiles from the global chemical transport model TM5 (Krol et al., 2005). Even though reflected infrared radiances are used, this approach allows for the retrieval of CO over bodies of water if clouds are present; in their absence, most of the incoming radiation is absorbed by the water. We have used, for any given day, TROPOMI data files from the most recent processor version available (01.01.00 to 01.04.00), either offline or reprocessed.

## 3 Methodology

We compare tropospheric total CO column retrievals from MOPITT and TROPOMI with respect to their colocated AirCore counterparts. Additional comparisons of colocated MOPITT and truncated AirCore vertical profiles were also performed. Colocation criteria required that observations from the two instruments involved were acquired within ≤12 h from each other and that their horizontal distance was ≤50 km, which are the same thresholds routinely used in MOPITT validations over land (e.g., Deeter et al., 2019).

Comparisons of remote sounder retrievals obtained with optimal estimation-based methods and *in situ* measurements must take into account the characteristics of the retrieval, e.g., its averaging kernels, or AK, and *a priori* (Rodgers and Connor, 2003). The MOPITT algorithm is based on optimal estimation as developed by Rodgers (2000); thus, for MOPITT

$$C_{ret} = C_a + A_c(X_{true} - X_a), \tag{1}$$

where $C_{ret}$ is the retrieved total column value, $C_a$ is the *a priori* total column value, $A_c$ is the total column averaging kernel, $X_{true}$

is the true profile value, and $X_a$ is the *a priori* profile value. $A_c$ is unitless, all other variables are expressed in column density, i.e., molecules per unit area. By applying Eq. 1 to the *in situ* profile, we can simulate the effects of the remote sounder retrieval and produce a 'smoothed' version of the true measurement which can, then, be directly compared to the sounder retrieval.





The Rodgers and Connor (2003) methodology is not applicable to TROPOMI retrievals, because the TROPOMI algorithm is not based on the optimal estimation method, but on Tikhonov regularization (Vidot et al., 2012; Borsdorff et al., 2014; Landgraf et al., 2016, and references therein). For TROPOMI

$$C_{ret} = A_c X_{true} \qquad (2)$$

Applying Eq. 2 to the *in situ* profile results in a retrieval-simulated (smoothed) version of the *in situ* measurement which can be directly compared to the TROPOMI retrieval.

Prior to obtaining smoothed AirCore total CO columns, complete (e.g., from the surface to the top of the atmosphere) AirCore CO profiles were generated following the standard method for MOPITT validation with aircraft data (Martínez-Alonso et al., 2020). AirCore profiles were interpolated to match the MOPITT *a priori* 35-level vertical grid, which preserves high vertical resolution in the troposphere. Empty levels at the bottom of each interpolated profile (levels with no CO value) were filled with the interpolated measurement closest to the surface. Empty levels between the top of the interpolated profile and the tropopause would usually be filled with the interpolated measurement closest to the tropopause; however, because all AirCore profiles reached the tropopause, this step was not necessary. Finally, empty levels above the tropopause were filled with colocated MOPITT *a priori* CO values. The now complete AirCore profiles were interpolated to match the 10-level vertical grid of the MOPITT retrievals. Total CO column values were derived from the vertical profiles as follows:

$$C = 2.12 * 10^{13} \sum_{i=1}^{n} \Delta p_i x_i \qquad (3)$$

where $C$ is the total column value in molec cm$^{-2}$, the constant *2.12*10$^{13}$* is in molec cm$^{-2}$ hPa$^{-1}$ ppb$^{-1}$, $n$ is the number of partial columns in the profile, $\Delta p_i$ is the thickness of partial column $i$ in hPa, and $x_i$ is the mean volume mixing ratio for the layer above level $i$ reported in ppb units. The derivation of Eq. (3) can be found in Deeter (2009).

Statistics values from the comparison of the satellite datasets (MOPITT, TROPOMI) with respect to AirCore *in situ* measurements were then calculated (satellite minus AirCore).

Additionally, we calculated the error introduced by approximating TROPOMI's partial columns below cloud top with the TROPOMI reference profiles by calculating the null-space error ($e_{null}$) of the TROPOMI retrieval process (Borsdorff et al., 2014; Landgraf et al., 2016):

$$e_{null} = (I - A_c) X_{true} \qquad (4)$$

where $I$ (a vector of ones) is the total column operator. This error is only important when total column TROPOMI AK are not used in retrieval comparisons or validations; otherwise, $e_{null}$ is, by definition, zero.

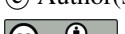



# 4 Results

## 4.1 MOPITT vs AirCore

CO values from the MOPITT version 8 and AirCore datasets were compared following the standard validation procedure as described above. Additionally, we performed an analogous comparison of MOPITT data with respect to the NOAA aircraft flask dataset traditionally used in MOPITT validation, for reference. To avoid ambiguities, we constrained both sets of comparisons to the period between January 2012 (the start of the AirCore dataset) and December 2019 (the most recent aircraft data officially available at the time of writing). Figure 4 summarizes CO bias values obtained in these comparisons; those values are also shown in Table 1, where full MOPITT validation results for the 2000-2018 period with respect to the aircraft dataset (from Deeter et al., 2019) are included for reference.

Biases between the three MOPITT variants (TIR, NIR, and multispectral) relative to AirCore are well below the MOPITT 10 % target accuracy (Francis et al., 2017) in all cases. MOPITT TIR partial column biases range from -2.4 % at 600 hPa and 0.9 % at 300 hPa; their mean is -0.85 %. MOPITT TIR total column bias is $0.0 \times 10^{17}$ molec cm$^{-2}$. MOPITT NIR partial column biases are even smaller, ranging between 0.2 % at 100 hPa and 1.3 % at 900, 800, 500, and 400 hPa, with a 1.1 % mean. The NIR total column bias is $0.3 \times 10^{17}$ molec cm$^{-2}$. Partial column biases for the MOPITT multispectral variant are between -5.8 % and 2.4 % at 500 and 300 hPa, respectively; the mean is -1.64 %. MOPITT multispectral total column bias is $0.1 \times 10^{17}$ molec cm$^{-2}$. Partial column standard deviation (SD) values range from 1.4 (for MOPITT TIR at 100 hPa) to 13.5 % (TIR+NIR, 300 hPa), with a mean of 6.9 %. For total column, the SD values are between 0.9 and $1.0 \times 10^{17}$ molec cm$^{-2}$ (mean $= 0.97 \times 10^{17}$ molec cm$^{-2}$).

Biases between the three MOPITT variants and aircraft profiles are also analyzed here for reference; next we describe results for the same time period covered by AirCore. MOPITT TIR partial column biases range from -1.7 % at 700 and 600 hPa and 3.9 % at 300 hPa, with a 0.3% mean. MOPITT TIR total column bias is $0.1 \times 10^{17}$ molec cm$^{-2}$. MOPITT NIR partial column biases range between 0.1 % at 100 hPa and 1.8 % at 900 hPa; mean is 0.84 %. MOPITT NIR total column bias is $0.3 \times 10^{17}$ molec cm$^{-2}$. MOPITT multispectral total column biases go from -5.6 % at 600 hPa to 7.8 % at 200 hPa, with a -0.23% mean. MOPITT multispectral total column bias is $0.2 \times 10^{17}$ molec cm$^{-2}$. Partial column SD values range from 1.6 % (NIR, 100 hPa) to 16.5 % (TIR+NIR, 300 hPa), with a mean of 7.5 %. For total column, the SD values are between 1.1 and $1.4 \times 10^{17}$ molec cm$^{-2}$ (mean $= 1.23 \times 10^{17}$ molec cm$^{-2}$). These statistical results are in good agreement with the values reported by Deeter et al. (2019) for the 2000-2018 period (Table 1).

MOPITT partial column biases with respect to both the AirCore and the aircraft datasets follow very similar vertical patterns (Fig. 4). The MOPITT multispectral variant displays the most extreme bias values. TIR and multispectral biases are close to zero near the surface, become negative between the surface and 500 hPa, and then positive between 300 and 100 hPa. In general, MOPITT biases with respect to AirCore and aircraft data are similar in the low-mid troposphere (i.e., between the surface and 500 hPa) for the TIR and multispectral variants. In contrast, biases for these two variants are closer to zero in the upper troposphere (300-100 hPa) when AirCore is involved. The NIR variant shows very small positive biases at all pressure levels for both AirCore and aircraft data.





### 4.1.1    Effect of extending shorter aircraft profiles upwards

As stated earlier, aircraft profiles used in MOPITT validation do not, in most cases, sample the entire troposphere due to limitations in the maximum altitude reachable by the sampling aircraft. The aircraft profiles in these cases are extended to the tropopause using the interpolated aircraft measurement closest to the tropopause and above the tropopause using *a priori* data from the CAM-chem model (Community Atmosphere Model with chemistry, Lamarque et al., 2012) for the same location and month of the aircraft profile to be extended. The error introduced in validation by extending aircraft profiles upwards was

expected to be small, but quantifying it had not been possible in the past due to the lack of suitable *in situ* measurements intrinsic to this problem. The AirCore dataset brings, for the first time, the opportunity to quantify this error. To this effect, we simulated a shorter AirCore dataset by truncating all AirCore profiles at 7000 m, i.e., slightly above the 400 hPa pressure threshold which must be reached by aircraft profiles to be usable in MOPITT validation. The truncated AirCore profiles were then extended upwards using the closest measurement to the tropopause and *a priori* CO data, and compared to the MOPITT

dataset as described earlier. For consistency, we constrained this analysis to the period between January 2012 and December 2019. Results are summarized in Table 2 and Fig. 4.

Biases between the three MOPITT variants and truncated AirCore are well below the MOPITT 10 % target accuracy. MOPITT TIR partial column biases range from -1.4 % at 600-700 hPa and 3.5 % at 300 hPa; their mean is 0.25 %. MOPITT TIR total column bias is 0.2 x $10^{17}$ molec cm$^{-2}$. MOPITT NIR partial column biases range between 0.4 % at 100 hPa and 2.1

% at 900 hPa, with a 1.73 % mean. The NIR total column bias is 0.4 x $10^{17}$ molec cm$^{-2}$. Partial column biases for MOPITT multispectral are between -4.9 % and 5.6 % at 600 and 300 hPa, respectively; the mean is -0.55 %. MOPITT multispectral total column bias is 0.2 x $10^{17}$ molec cm$^{-2}$. Partial column SD values range from 1.4 % (NIR, 100 hPa) to 11.6 % (TIR+NIR, 300 hPa), with a mean of 6.2 %. For total column, the SD values are between 0.8 and 0.9 x $10^{17}$ molec cm$^{-2}$ (mean = 0.84 x $10^{17}$ molec cm$^{-2}$).

Figure 4 shows that biases between MOPITT and truncated AirCore partial columns differ from the MOPITT/AirCore biases described in Section 4.1. In general, biases between all three MOPITT variants and truncated AirCore profiles appear to shift to the right (i.e., increase slightly) with respect to the MOPITT/AirCore biases. For MOPITT TIR that results in biases that increase mostly in the upper troposphere, by up to 2.6 p.p. (percent points) at 300 hPa, and mimic very closely in sign and magnitude those between MOPITT and the aircraft data. For MOPITT NIR, biases increase almost uniformly at all pressure

levels, by 0.2-0.8 p.p. For MOPITT multispectral the change in bias is larger (up to 3.2 p.p. at 300 hPa), mimicking once more the MOPITT/aircraft biases. For total CO column the biases between MOPITT and truncated AirCore increase by 0.20, 0.10, and 0.10 x $10^{17}$ molec cm$^{-2}$ (for the TIR, NIR, and multispectral variants, respectively).

### 4.2    TROPOMI vs AirCore

TROPOMI retrieves total CO column values from solar reflected radiances over land (under clear and cloudy conditions)

and water (cloudy only). During TROPOMI retrieval parameters describing the cloud contamination of the measurement and interfering trace gases are quantified together with the CO column. The reference CO profiles used in the retrieval are scaled



based on estimated above-cloud CO rather than based on estimated total CO (Landgraf et al., 2016). A previous comparison of clear MOPITT and TROPOMI total CO column retrievals showed good agreement between the two datasets (Martínez-Alonso et al., 2020). In that same study, cloudy TROPOMI CO retrievals over bodies of water were also validated with respect to ATom

(Atmospheric Tomography mission; Wofsy et al., 2018) aircraft profiles. The results showed that the $e_{null}$ (null-space error) of the profile scaling retrieval over water is very small (2.16 %). The authors concluded that, since there are no major emission sources over water, CO values closer to the surface (most likely to be below cloud top) are well characterized by the scaled reference profiles. Larger errors could occur, however, in cloudy TROPOMI land retrievals, particularly near CO emission sources, if not accounting for the TROPOMI AK. Their analysis could not be extended over land because the ATom campaign

was designed to sample the troposphere mostly over oceans. Here we extend the Martínez-Alonso et al. (2020) analysis by characterizing the error introduced by clouds in land TROPOMI CO retrievals using CO profiles from the AirCore dataset.

We analyzed separately TROPOMI clear and cloudy data. TROPOMI clear-sky and clear-sky like observations are defined by aerosol optical thickness < 0.5 and cloud altitude values < 500 m; they correspond to TROPOMI quality assurance (QA) value = 1.0. TROPOMI observations with mid-level clouds are those with aerosol optical thickness $\geq$ 0.5 and cloud altitude

values < 5000 m; QA = 0.7 (Landgraf et al., 2021). Comparisons of clear/cloudy TROPOMI total CO column values with respect to their colocated AirCore counterparts are summarized in Fig. 5 and Table 3. Under clear conditions, TROPOMI has similarly low bias values (1.27 % and 1.61 %) with respect to both true and smoothed AirCore total CO column values; the latter account for TROPOMI vertical sensitivity to CO as shown in Eq. 2. The Pearson correlation coefficient (R) values (0.81 and 0.82) indicate a slight improvement in the fit when the AK are applied. The slope of the fitted line remains unchanged

(0.96). Under cloudy conditions the change in biases is also small (1.03 % and 2.02 % for true and smoothed AirCore values, respectively); the R values (0.74 and 0.76) and slope of the linearly fitted line (0.80 and 0.83) show larger improvement of the fit when the TROPOMI AK are accounted for. Figure 6 shows that, overall, the distribution of bias values is mostly symmetrical with respect to the zero % bias value, i.e., relative biases show no obvious latitudinal dependence, although the latitudinal coverage of available AirCore data is limited.

The TROPOMI null-space error ($e_{null}$) is indicative of differences between the shape of TROPOMI CO reference profiles and that of true CO profiles which may result in differences between the true and TROPOMI-retrieved total CO column values. We have calculated $e_{null}$ values between TROPOMI and AirCore profiles over land using Eq. 4; results as a function of latitude are shown in Fig. 7. Under clear conditions, the TROPOMI total column averaging kernel $A_c$ closely matches the total column operator $I$ such that, according to Eq. 4, the null-space error $e$ is close to zero. The relative mean and SD values of $e_{null}$ in this

case are 0.36 % $\pm$ 0.66 %, or 0.61 $\pm$ 1.14 x $10^{16}$ molec cm$^{-2}$ (Fig. 7.a). In cloudy conditions TROPOMI is more sensitive to CO above the clouds than to CO below them; in these cases if the shape of the TROPOMI reference profiles does not properly represent that of the actual CO profiles a null-space error is introduced. Our results indicate that the relative mean and SD values of $e_{null}$ are in this case slightly larger: 0.98 % $\pm$ 2.32 %, or 1.65 $\pm$ 4.15 x $10^{16}$ molec cm$^{-2}$ (Fig. 7.b).



## 5   Discussion

Here we have evaluated the repeatability and biases of *in situ* AirCore CO vertical profiles. We have compared the AirCore profiles to the MOPITT version 8 CO dataset to assess their performance in validation efforts and to quantify errors introduced in validation by the use of CO vertical profiles lacking upper tropospheric *in situ* measurements, a common issue in aircraft datasets. Finally, we have used AirCore data to estimate the error introduced by clouds in TROPOMI land CO retrievals.

From CO profiles acquired by pairs of AirCore systems deployed simultaneously and from the same site we have estimated

that the average repeatability at most altitudes is well below ± 2 ppb (Fig. 2). Our analysis shows lower repeatability values (5-15 ppb) between 50 and 70 hPa consistent with higher uncertainty in AirCore stratospheric CO retrievals attributable to AirCore surface effects, chemical interactions or measurement interferences from other trace gas species, or incorrect AirCore sample end-member assumptions (Chen et al., 2022, in prep.). Colocated (<2 hours and <15 km apart) CO profiles from AirCore and NOAA aircraft flask profiles indicate that AirCore biases are between -6 and +6 ppb, with a 0.6 ppb overall average bias (Fig.

3.a).

Our MOPITT comparisons show that AirCore provides validation results analogous in magnitude and sign to those from the NOAA aircraft flask dataset (Fig. 4); biases are in all cases well below the MOPITT 10 % target accuracy (Francis et al., 2017). MOPITT/AirCore and MOPITT/aircraft biases between the surface and 500 hPa differ very slightly (by < 0.5 p.p. on average for all MOPITT variants). The same is true for MOPITT NIR/AirCore biases at all pressure levels. Between 400

and 200 hPa, though, AirCore is closer to MOPITT than the aircraft dataset is by (in average) 2.7 p.p. (TIR) and 4.7 p.p. (multispectral). Larger biases between MOPITT and the aircraft dataset at that pressure range are consistent with the fact that profiles from the latter do not reach in most cases above 400 hPa and have to be extended upwards. In contrast, the low MOPITT/AirCore biases indicate good agreement between the two datasets and imply that previous validation results may have overestimated the magnitude of MOPITT retrieval biases in that upper tropospheric region. Further up in the troposphere,

at 100 hPa, both MOPITT/AirCore and MOPITT/aircraft biases approximate zero for all MOPITT variants. Because MOPITT profiles are less sensitive to CO at/above 100 hPa, the *a priori* dominates retrievals at that pressure level, leading to low biases (both MOPITT/AirCore and MOPITT/aircraft biases) due to cancellation of the dominant *a priori* term in the difference of retrieved and smoothed *in situ* profiles.

In order to investigate the effects of extending upwards shorter tropospheric aircraft CO profiles used in validation, we

have simulated a truncated version of the AirCore CO dataset which we have compared to MOPITT retrievals. Differences between MOPITT/AirCore and MOPITT/truncated-AirCore biases (Fig. 4) are small, < 1 p.p. on average. We observe that, for the TIR and multispectral variants, MOPITT/truncated-AirCore biases depart from MOPITT/AirCore biases to mimic the MOPITT/aircraft biases. These results reinforce our interpretation regarding previous validation efforts having slightly overestimated the magnitude of MOPITT retrieval biases in the upper troposphere due to the use of shorter tropospheric aircraft

CO profiles. While always small, the effects are relatively stronger (2 to 3 p.p.) at 400-200 hPa; more modest effects can also be seen at other pressure levels. This is because at any given pressure level $P$ the MOPITT CO retrievals are not only sensitive to CO at that level, but to CO at other levels too. That is, the MOPITT AK (with which the AirCore profiles are convolved





prior to bias calculations) are not delta functions peaking at level $P$, but curves of increasing amplitude towards level $P$. We also observe that, for the NIR variant, the effects are similar at all pressure levels. This is consistent with the MOPITT NIR

retrievals being sensitive to total CO column only, i.e., the MOPITT NIR AK are not curve-like, but rather flat. Our findings support results by Tang et al. (2020), where *in situ* aircraft profiles extended with reanalysis data were compared to MOPITT multispectral retrievals. The authors found good agreement between MOPITT and the extended aircraft profiles at the surface layer; at upper levels (400 and 200 hPa), biases increased due to limited aircraft observations.

Finally, we have used the AirCore dataset to investigate cloud effects in TROPOMI total CO column retrievals over land.

The mean relative bias between clear TROPOMI and smoothed AirCore (1.61 %) is only slightly smaller than that between cloudy TROPOMI and smoothed AirCore (2.02 %) (Fig. 5.b and 5.d). Mean relative biases between TROPOMI and true (i.e., unsmoothed) AirCore are 1.27 % and 1.03 % (for clear and cloudy TROPOMI retrievals, respectively); we note that, although both are very small and differ by only 0.24 p.p., the mean bias is higher for clear observations. However, the other quality-of-fit indicators (R and linear fit slope) show that TROPOMI CO retrievals are closer in value to true AirCore CO under clear

conditions. Borsdorff et al. (2018) reported a similarly small difference in bias (0.2 ppb, equivalent to ~0.25 p.p.) between clear and cloudy TROPOMI CO observations with respect to *in situ* ground measurements over 9 remote sites. Our results indicate that TROPOMI/AirCore biases for cloudy observations over land do not show obvious latitudinal effects (Fig. 6).

The null-space error ($e_{null}$) quantifies the expected difference between the true CO column and the retrieved TROPOMI CO column due to differences between the shape of the true profile and that of the TROPOMI reference profile. It is only

relevant when the TROPOMI CO retrievals are compared with respect to other reference measurements whitout accounting for the sensitivity loss caused by clouds; $e_{null}$ can be completely avoided by using the TROPOMI total column AK provided in the data product. Our null-space error calculations using AirCore CO data show that the magnitude of the error introduced in cloudy TROPOMI CO retrievals over land by using scaled reference profiles is very small (0.98 % $\pm$ 2.32 %, or 1.65 $\pm$ 4.15 x $10^{16}$ molec cm$^{-2}$), and slightly skewed towards positive values (Fig. 7.b). These observations are in agreement with

results reported by Martínez-Alonso et al. (2020) in their analysis of cloudy CO observations from TROPOMI and ATom-4 over bodies of water (relative mean and SD values 2.16 % $\pm$ 2.23 %, or 3.70 $\pm$ 3.75 x $10^{16}$ molec cm$^{-2}$). The prevalence of positive null-space error values suggests that, on average, the TROPOMI reference profiles analyzed may have too much CO near the surface, and thus result in TROPOMI retrievals which may overestimate the below-cloud partial column. No latitudinal dependence was observed in the null-space error values either in this analysis nor in the Martínez-Alonso et al. (2020) study.

**6 Conclusions**

AirCore is a novel airborne sampler suited for the validation of satellite retrievals of tropospheric CO and, potentially, other relevant tropospheric gases and parameters such as $CO_2$, $CH_4$, temperature, and relative humidity because, unlike most aircraft platforms, it samples continuously from the lower stratosphere to near the surface. According to our analysis, the mean bias (with respect to the NOAA aircraft flask dataset) and repeatability of CO AirCore measurements are near 0.6 and 0.03 ppb,

respectively. AirCore measurements from near the surface are currently being discarded because they are affected by end-



member mixing with spiked CO push gas; higher stratospheric uncertainties of up to ~15 ppb (this study; Chen et al., 2022, in prep.) have also been identified. Efforts are being made towards solving these problems in the near future.

We have validated a temporal subset (2012-2019) of the MOPITT version 8 data with respect to AirCore profiles by applying the procedure used in previous MOPITT validation efforts with respect to aircraft *in situ* measurements (Deeter et al., 2019, and references therein). As a reference, we have also validated the same MOPITT temporal subset with respect to NOAA aircraft flask profiles. The resulting MOPITT/AirCore and MOPITT/aircraft biases are very similar and align well with the full MOPITT validation results reported by Deeter et al. (2019).

We find MOPITT/AirCore biases at 400-200 hPa to be smaller than their MOPITT/aircraft counterparts; it is also at that pressure range that MOPITT/AirCore and MOPITT/truncated-AirCore biases differ the most. Both pieces of evidence indicate that extending upwards shorter aircraft profiles (i.e., aircraft profiles that sample up to the required 400 hPa MOPITT validation threshold but not above it) results in small validation errors in the upper troposphere (up to 2-3 p.p. in the 400-200 hPa range) and, thus, in a slight overestimation of MOPITT retrieval biases in that region.

Our TROPOMI/AirCore analysis shows that the TROPOMI approach to retrieve total CO column values under cloudy conditions results in small biases over land (1-2 %); similarly small biases over bodies of water had been previously reported by Martínez-Alonso et al. (2020). We must keep in mind, however, that this study's results may be representative of unpolluted areas only. AirCore *in situ* measurements are commonly performed away from CO emission sources such as heavily populated areas, industrial regions, or active fires, where CO concentrations at the boundary layer (and, thus, most likely to be below cloud-top) would be more variable and, thus, could depart from the TROPOMI reference profile values. AirCore measurements near CO emission sources would be needed to fully evaluate the TROPOMI approach and the performance of its reference profiles. Our null-space error calculations show that the magnitude of the error introduced in cloudy TROPOMI retrievals over land by using scaled reference profiles is very small (~0.98 %), does not show latitudinal dependencies, and is slightly skewed towards positive values. While the AirCore dataset spans a ~10 year time frame, it is still rather limited geographically; more latitudinally widespread measurements are needed to study whether there are substantial latitudinal dependencies in the TROPOMI retrievals.

*Data availability.* AirCore data from the 13 August 2021 version are publicly available from the NOAA Global Monitoring Laboratory upon request https://doi.org/10.15138/6AV0-MY81. NOAA aircraft flask data version 2.0 from the 9 February 2021 version were obtained from https://gml.noaa.gov/ccgg/obspack/data.php (Sweeney et al., 2021). MOPITT data from version 8 can be downloaded from https://doi.org/10.5067/TERRA/MOPITT/MOP02T_L2.008 (Ziskin, 2019c) (TIR), https://doi.org/10.5067/TERRA/MOPITT/MOP02N_L2.008 (Ziskin, 2019b) (NIR), and https://doi.org/10.5067/TERRA/MOPITT/MOP02J_L2.008 (Ziskin, 2019a) (TIR + NIR). TROPOMI level 2 CO retrievals for 7 November 2017 to 27 June 2018 were downloaded from https://s5pexp.copernicus.eu/ (last access: 27 November 2019), (ESA, 2018a); retrievals for dates after 28 June 2018 were downloaded from https://s5phub.copernicus.eu/ (last access: 9 February 2021), (ESA, 2018b).



*Author contributions.* SMA defined the concept and methodology of the paper, performed the formal analysis of data sets, software development, and data presentation. MND provided additional software for MOPITT validation. SMA, MND, and HW oversaw the MOPITT data

analysis. TB and IA oversaw the TROPOMI data analysis. BCB and CS oversaw the AirCore data analysis. KM and CS oversaw the NOAA aircraft flask data analysis. SMA wrote the original draft. All authors contributed to the review and editing of this paper.

*Competing interests.* Some authors are members of the editorial board of the Atmospheric Measurement Techniques journal. The peer-review process was guided by an independent editor, and the authors have also no other competing interests to declare.

*Acknowledgements.* NCAR internal reviews provided by Gene Francis and Wenfu Tang are greatly appreciated. Sentinel-5 Precursor is part of the EU Copernicus program, and Copernicus (modified) Sentinel data for 2017–2021 have been used.

*Financial support.* This material is based upon work supported by the National Center for Atmospheric Research (NCAR), which is a major facility sponsored by the National Science Foundation (grant no. 1852977). The NCAR MOPITT project is supported by the National Aeronautics and Space Administration (NASA) Earth Observing System (EOS) Program.





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




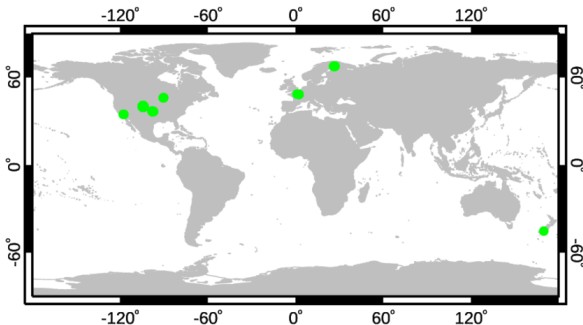

**Figure 1.** AirCore vertical profile locations listed from west to east. USA: Edwards Air Force Base (California), Boulder (Colorado), Lamont (Oklahoma), and Park Falls (Wisconsin). France: Traînou. Finland: Sodankylä. New Zealand: Lauder.



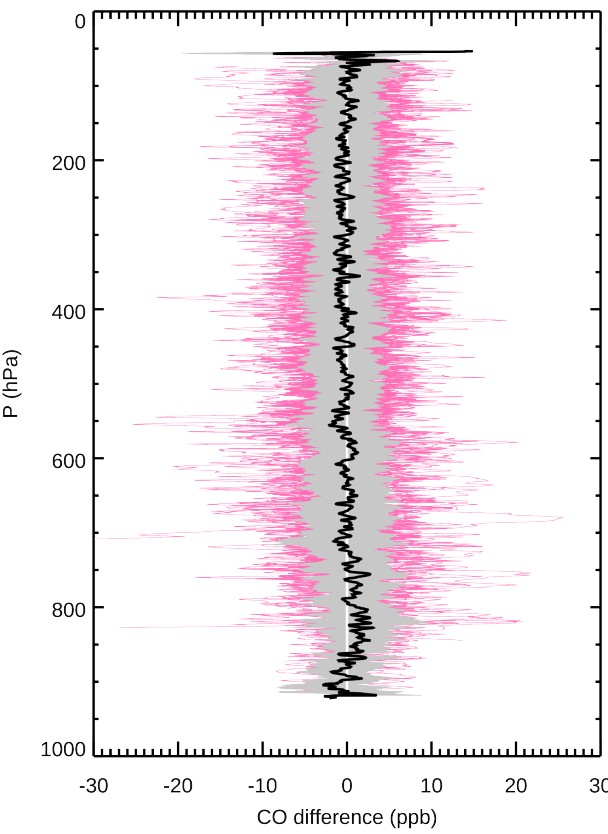

**Figure 2.** Differences between CO vertical profiles acquired by pairs of AirCore systems deployed simultaneously and from the same location. Pink: biases for each AirCore pair. Black: mean of all biases. Gray: mean ±1 standard deviation (SD).





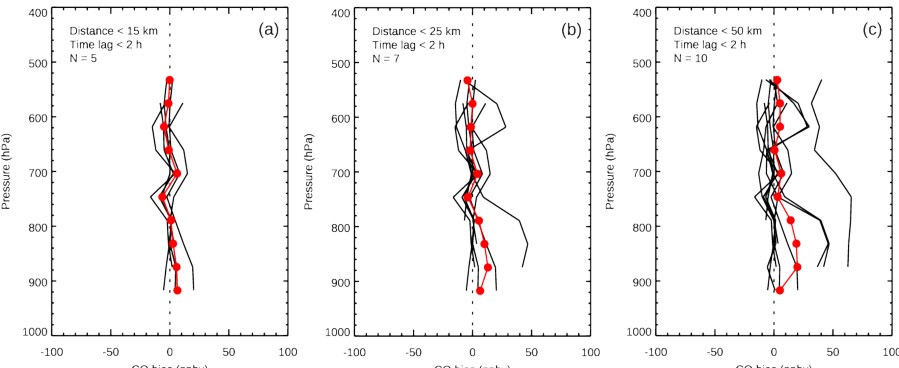

**Figure 3.** Differences between colocated CO vertical profiles from AirCore and NOAA aircraft flask campaign (AirCore minus aircraft data). Black lines show bias for each colocated pair, averaged biases from all pairs are shown in red. Colocation criteria and number of colocated pairs are indicated in each panel.





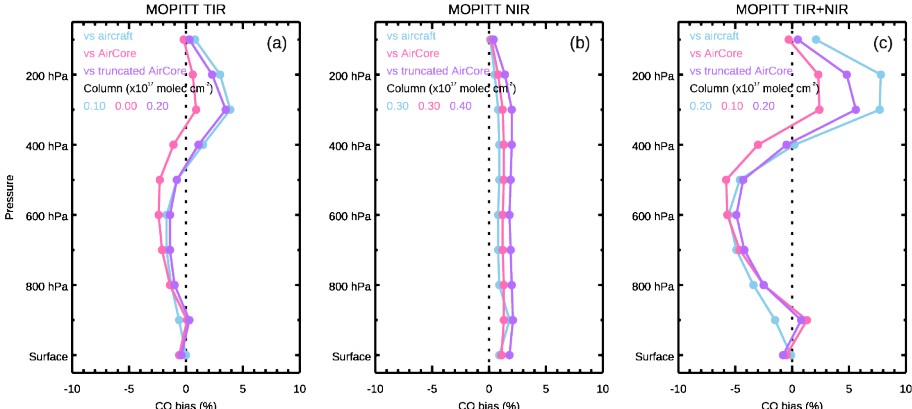

**Figure 4.** CO biases for the 2012-2019 period from the comparison of MOPITT with respect to NOAA aircraft flask data (blue), AirCore profiles (pink), and truncated AirCore profiles extended upwards (purple). (a) For MOPITT TIR. (b) For MOPITT NIR. (c) For MOPITT multispectral. Relative bias in %. Column bias in units of $10^{17}$ molec cm$^{-2}$. The $\pm10$ % CO bias range is equal to the MOPITT target accuracy (Francis et al., 2017).



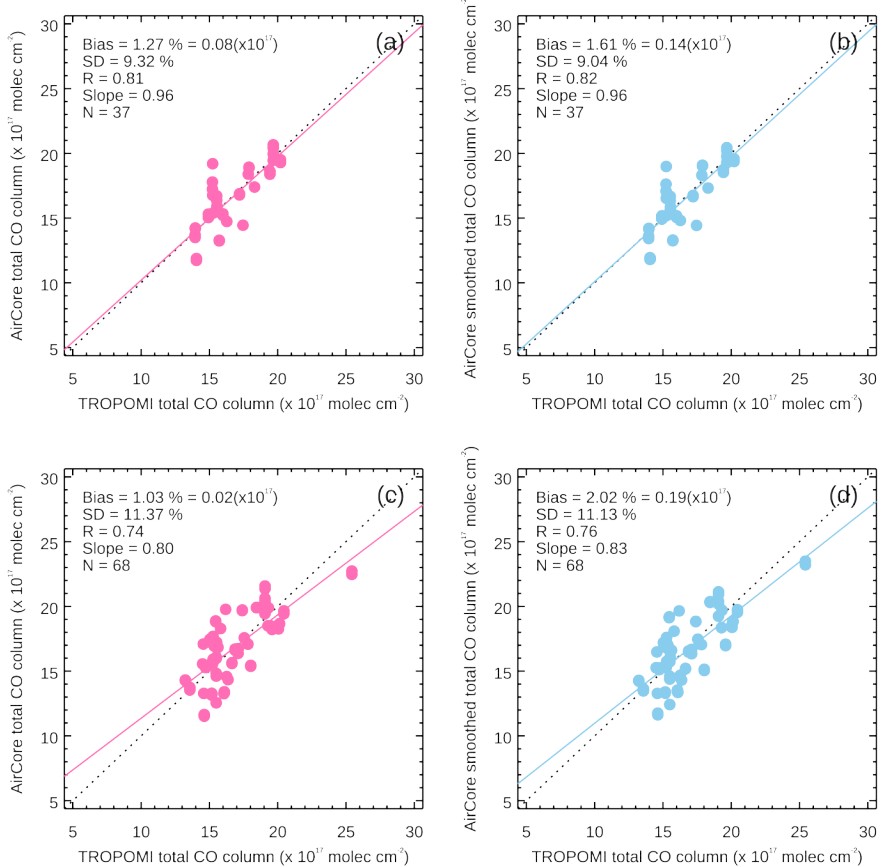

**Figure 5.** Comparison of total CO column values from TROPOMI and AirCore for the November 2017-July 2021 period. Top row panels (a) and (b) are both for TROPOMI clear-sky and clear-sky like observations (i.e., QA = 1.0). Bottom row panels (c) and (d) are for TROPOMI observations with mid-level clouds (i.e., QA = 0.7). Left column panels (a) and (c) show true AirCore data. Right column panels (b) and (d) show smoothed AirCore data to account for TROPOMI vertical sensitivity to CO. Bias values shown both as percentage (%) and in units of molec cm$^{-2}$.





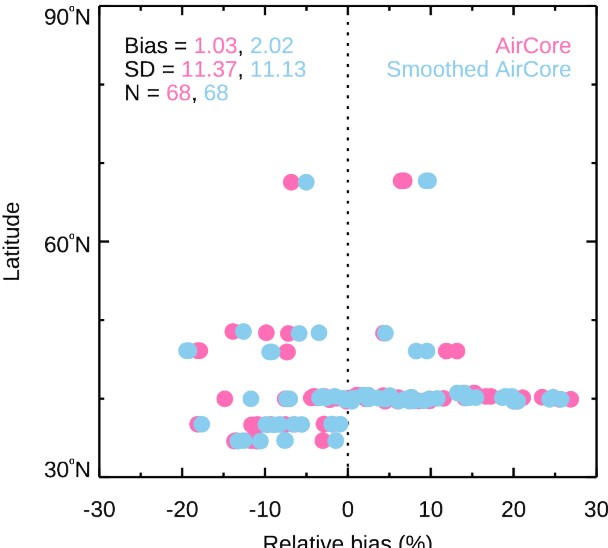

**Figure 6.** Latitudinal distribution of bias values between TROPOMI and AirCore cloudy observations over land.





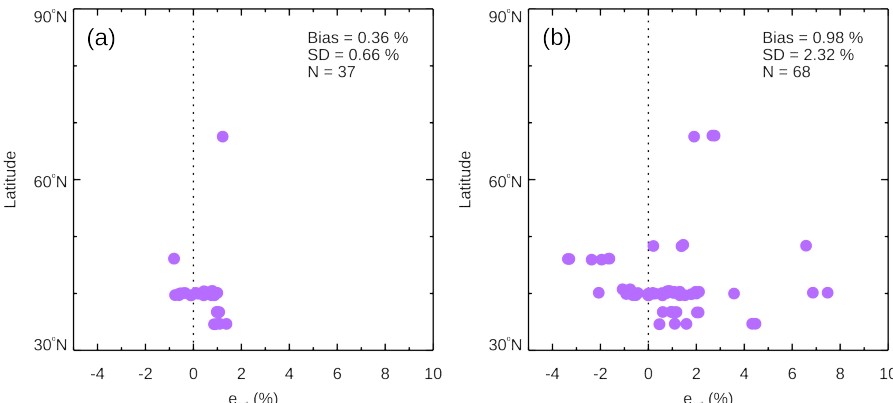

**Figure 7.** Latitudinal distribution of null-space error between TROPOMI and AirCore observations over land. (a) For clear observations. (b) For cloudy observations.



**Table 1.** Statistics from the comparison of MOPITT CO with respect to AirCore and to NOAA aircraft flask data for the 2012-2019 period. Statistics from the validation of MOPITT CO with respect to aircraft data for 2000-2018 (Deeter et al., 2019) are also included for reference. Column bias and SD in units of $10^{17}$ molec cm$^{-2}$. Partial column relative bias and SD in %. Partial column results only shown for even pressure levels, for simplicity.

| | | Total Column | Surface | 800 hPa | 600 hPa | 400 hPa | 200 hPa |
|---|---|---|---|---|---|---|---|
| MOPITT TIR vs AirCore | Bias | 0.0 | -0.6 | -1.4 | -2.4 | -1.1 | 0.6 |
| (2012-2019) | SD | 0.9 | 4.3 | 5.7 | 6.7 | 10.0 | 6.9 |
| | r | 0.71 | 0.82 | 0.74 | 0.63 | 0.52 | 0.49 |
| MOPITT TIR vs aircraft | Bias | 0.1 | 0.0 | -1.3 | -1.7 | 1.5 | 3.0 |
| (2012-2019) | SD | 1.2 | 5.1 | 5.8 | 6.8 | 9.8 | 7.5 |
| | r | 0.84 | 0.80 | 0.83 | 0.83 | 0.74 | 0.56 |
| MOPITT TIR vs aircraft | Bias | 0.2 | 0.5 | -0.7 | -1.3 | 1.6 | 3.0 |
| (2000-2018) | SD | 1.4 | 5.7 | 7.2 | 8.3 | 11.2 | 8.3 |
| | r | 0.82 | 0.74 | 0.77 | 0.80 | 0.72 | 0.54 |
| MOPITT NIR vs AirCore | Bias | 0.3 | 1.1 | 1.3 | 1.2 | 1.3 | 0.8 |
| (2012-2019) | SD | 1.0 | 4.6 | 5.4 | 5.4 | 5.8 | 4.0 |
| | r | 0.30 | 0.42 | 0.37 | 0.32 | 0.32 | 0.46 |
| MOPITT NIR vs aircraft | Bias | 0.3 | 0.9 | 0.9 | 0.8 | 0.9 | 0.5 |
| (2012-2019) | SD | 1.1 | 5.5 | 5.8 | 5.7 | 6.0 | 4.5 |
| | r | 0.57 | 0.57 | 0.61 | 0.62 | 0.62 | 0.60 |
| MOPITT NIR vs aircraft | Bias | 0.1 | 0.1 | -0.1 | -0.2 | -0.1 | -0.4 |
| (2000-2018) | SD | 1.3 | 6.3 | 6.5 | 6.2 | 6.6 | 4.8 |
| | r | 0.60 | 0.60 | 0.62 | 0.64 | 0.64 | 0.61 |
| MOPITT TIR+NIR vs AirCore | Bias | 0.1 | -0.5 | -2.5 | -5.7 | -3.0 | 2.3 |
| (2012-2019) | SD | 1.0 | 8.1 | 8.6 | 8.8 | 10.3 | 11.5 |
| | r | 0.73 | 0.67 | 0.63 | 0.46 | 0.43 | 0.16 |
| MOPITT TIR+NIR vs aircraft | Bias | 0.2 | -0.1 | -3.4 | -5.6 | 0.2 | 7.8 |
| (2012-2019) | SD | 1.4 | 9.9 | 9.6 | 8.6 | 12.4 | 13.4 |
| | r | 0.83 | 0.66 | 0.73 | 0.80 | 0.66 | 0.30 |
| MOPITT TIR+NIR vs aircraft | Bias | 0.2 | -0.1 | -2.7 | -5.1 | 0.2 | 6.7 |
| (2000-2018) | SD | 1.6 | 9.8 | 11.7 | 10.6 | 14.1 | 14.7 |
| | r | 0.81 | 0.62 | 0.68 | 0.76 | 0.64 | 0.30 |





**Table 2.** Statistics from the comparison of MOPITT CO with respect to truncated AirCore profiles extended upwards. Column bias and SD in units of $10^{17}$ molec cm$^{-2}$. Partial column relative bias and SD in %. Partial column results only shown for even pressure levels, for simplicity.

|  |  | Total Column | Surface | 800 hPa | 600 hPa | 400 hPa | 200 hPa |
|---|---|---|---|---|---|---|---|
| MOPITT TIR vs truncated AirCore | Bias | 0.2 | -0.4 | -1.0 | -1.4 | 1.1 | 2.3 |
| (2012-2019) | SD | 0.8 | 4.0 | 5.0 | 5.5 | 8.5 | 6.1 |
|  | r | 0.78 | 0.84 | 0.78 | 0.73 | 0.67 | 0.63 |
| MOPITT NIR vs truncated AirCore | Bias | 0.4 | 1.8 | 2.0 | 1.8 | 2.0 | 1.4 |
| (2012-2019) | SD | 0.9 | 4.4 | 5.1 | 5.0 | 5.3 | 3.8 |
|  | r | 0.38 | 0.46 | 0.42 | 0.39 | 0.39 | 0.49 |
| MOPITT TIR+NIR vs truncated AirCore | Bias | 0.2 | -0.8 | -2.5 | -4.9 | -0.5 | 4.8 |
| (2012-2019) | SD | 0.8 | 8.0 | 8.5 | 7.7 | 6.9 | 11.0 |
|  | r | 0.82 | 0.68 | 0.64 | 0.56 | 0.77 | 0.34 |





**Table 3.** Summary of statistics from the comparison of total CO column values from TROPOMI (under either clear or cloudy conditions) and AirCore. Bias values are provided in percent (%) and in units of $10^{17}$ molec cm$^{-2}$ (in parentheses). SD values are shown in %.

|  |  | Total Column | Total Column 'smoothed' |
|---|---|---|---|
| TROPOMI vs AirCore | Bias | 1.27 (0.08) | 1.61 (0.14) |
| clear | SD | 9.32 | 9.04 |
|  | r | 0.81 | 0.82 |
| TROPOMI vs AirCore | Bias | 1.03 (0.02) | 2.02 (0.19) |
| cloudy | SD | 11.37 | 11.13 |
|  | r | 0.74 | 0.76 |