# Peer review of "Evaluation of MOPITT and TROPOMI carbon monoxide retrievals using AirCore *in situ* vertical profiles"

_Atmospheric Measurement Techniques, 2022_

## Author Comment (AC1)

**Responses to Review #1**

We appreciate very much your comments on this manuscript, please find our responses below. (Line numbers are those in the original version of the manuscript.)

The authors use the advanced AirCore profiles operated by NOAA during the last years to validate the MOPITT and TROPOMI CO products. With the high measurement altitude of the AirCore measurements, the authors quantify the error introduced in MOPITT CO validation by the use of shorter aircraft vertical CO profiles extended upwards. The results are promising, and the error is estimated to be small. The AirCore profiles are also applied to validate the TROPOMI CO under both clear and cloudy conditions. The results are also consistent with previous studies. In general, the paper is well-written and easy to understand. I would like to recommend it to publish on AMT after addressing some minor comments/suggestions below:

lines 5-6 the unit is in mole cm-2 for MOPITT-AirCore, while in line 19 the unit is in % for TROPOMI-AirCore. Please be consistent, better to use %.

Thank you for catching this inconsistency. We have modified the abstract so MOPITT bias and SD values are now in %:
"Mean MOPITT/AirCore total column bias values and their standard deviation (0.4±5.5, 1.7±5.6, and 0.7±6.0 for MOPITT thermal-infrared, near-infrared, and multispectral retrievals, respectively; all in %) are similar to results obtained in MOPITT/NOAA aircraft flask data comparisons from this study and from previous validation efforts."

We have also added mean and SD of biases in % to Tables 1 and 2, and reworded the captions accordingly.

Same for the total column mean column bias values shown in Fig 4: we have added % values (only values in molec cm$^{-2}$ were shown before). We have reworded the caption accordingly.

Finally, for completeness, we have added to Fig. 5 standard deviation values in molec cm$^{-2}$ and to Table 3 standard deviation values in molec cm$^{-2}$ (before they were all in % only).

Line 88 – 91: it is not so clear for me to understand the uncertainty of the CO. The authors said that "the total uncertainty is typically <5 ppb (Karion et al., 2013)", but they also pointed out that "stratospheric CO profiles have shown differences up to ~15 ppb". Besides the uncertainty of the CO at each altitude, it is more important to highlight the uncertainty of the CO total column (or within the AirCore measurement vertical range)

We have determined that the AirCore-derived total CO column error is ~1.3 ppb. This considers natural variability and measurement errors due to instrument and the AirCore sampling system between balloon flights spaced 15 minutes apart in the lower 80% of the atmospheric column. We assume errors of 15 ppb above this level based on other studies. As it turns out, AirCores flown on the same balloon have significantly less variability than those flown 15 minutes apart, suggesting that the majority of the uncertainty in the AirCore-derived XCO is due to natural variability and not uncertainties in the AirCore sampling or measurement systems.

Because we don't have both the actual and expected total CO column values to compute percent error, we estimate it as follows:

sigma(XCO) / AirCore-calculated XCO = 1.3 ppb XCO error / 71 ppb XCO= ~1.8%.

The manuscript has been reworded as follows:

"CO in AirCore samples is measured by cavity ringdown-spectroscopy (CRDS) at a precision typically less than 5 ppb (Karion et al., 2013) for ~ 0.5 Hz measurements. AirCore CO is, however, still considered a developmental product due to its use for correcting end-member mixing in other trace gas profiles. Comparisons of stratospheric AirCore CO profiles have sometimes shown differences up to ~15 ppb, which could be a result of AirCore tubing surface interactions or diffusion effects. It is also possible that chemical interactions or measurement interferences from other trace gas species or incorrect AirCore sample end-member assumptions have been made. Given these uncertainties and the number of independent CO measurements in each AirCore sample, we derive an estimated AirCore XCO uncertainty of ~1.3 ppb (2 sigma), equivalent to ~1.8 %."

Figure 6, shows that the relative bias at one latitude (or at the same location) can vary from -18% to 28%. Have you ever investigated the cause for this, e.g. AirCore-satellite distance? cloud? surface? Meteorology?
We have not investigated this point in particular, but other results presented in the manuscript may help.

The AirCore/aircraft comparison results (Fig. 3) suggest that differences in the actual CO concentrations observed by each of the two instruments could be the cause. Biases are smaller when the instruments are closer in space and time (Fig. 3a, <15 km and <2 h) and increase as the distance between the instrument increases (Fig. 3b ad 3.c; <25 km and <50 km, respectively). Similarly, CO concentrations could differ between colocated AirCore and TROPOMI observations, which may be up to 12 h and 50 km apart.

We quantified the error introduced by clouds in land TROPOMI retrievals, which is lower (~2 % in average) than the spread mentioned in this comment.

For clarity, we added the following at the end of line 326: "The spread in biases shown in this figure may reflect differences in the actual CO concentrations observed by each of the two instruments, which may be up to 12 h and 50 km apart."

---

## Author Comment (AC2)

**Responses to Review #2**

We thank you for reviewing this manuscript, please find our responses below. (Line numbers are those in the original version of the manuscript.)

This is a well written paper with an important additional validation of MOPITT and TROPOMI CO retrievals. The main addition to previous validation studies is that the validation is more robust at higher altitudes due to the better vertical range covered by AirCore. I can recommend the paper for publication in AMT after a few minor changes. These are mainly related to the description of the errors, as detailed below.

Abstract. l. 12, 19 and 20: I find these percentage values difficult to understand as they do not make clear to what they refer (percentage of what?). Further e_null (l.20) is not explained and will not mean much to the non-expert reader, certainly in the abstract.
Line (l.) 12 shows percent points difference, that is, the difference between two percentage values, in this case between MOPITT-AirCore biases and MOPITT-truncated AirCore biases. For clarity, we have added a few words to the previous sentence, which now reads as follows: "Here we quantify for the first time the error introduced in MOPITT CO validation by the use of shorter aircraft vertical profiles extended upwards by analyzing validation results of MOPITT with respect to full and truncated AirCore CO vertical profiles."
l. 19 shows bias values between AirCore *in situ* measurements and TROPOMI retrievals, expressed as percent values. The AirCore *in situ* measurement are the reference and, thus, the percent is calculated as (100.*(TROPOMI-AirCore))/AirCore. For clarity, we have added "as the reference" to l. 17, which now reads: "We use AirCore measurements as the reference to evaluate the error [...]"
l. 20: e_null is introduced in lines 19 and 20 as follows: "the null-space error, which accounts for differences between the shape of TROPOMI reference profiles and that of AirCore true profiles". Additional explanations on the calculation and meaning of e_null are provided in the Methodology, Results, and Discussion sections.

l. 21: please include atmospheric before oxidation
We added "atmospheric" to l. 23, which now reads: "Tropospheric CO (carbon monoxide) is mostly produced by incomplete combustion of fuels, biomass burning, and atmospheric oxidation of CH4 (methane) and other hydrocarbons"

l 26. include reference for the statement on radiative forcing
We included this reference:
Szopa, S., Naik, V., Adhikary, B., Artaxo, P., Berntsen, T., Collins, W., Fuzzi, S., Gallardo, L., Kiendler-Scharr, A., Klimont, Z., Liao, H., Unger, N., and Zanis, P.: Short-Lived Climate Forcers, in Climate Change 2021: The Physical Science Basis. Contribution of Working Group I to the Sixth Assessment Report of the Intergovernmental Panel on Climate Change, Cambridge University Press, Cambridge, United Kingdom and New York, NY, USA, https://doi.org/10.1017/9781009157896.008, 2021.

l. 37. please include reference.
We included these references, where the method used to extend upwards the measured aircraft profiles to allow for comparison to satellite retrieved values:
Martínez-Alonso, S., Deeter, M. N., Worden, H. M., Gille, J. C., Emmons, L. K., Pan, L. L., Park, M., Manney, G. L., Bernath, P. F., Boone, C. D., Walker, K. A., Kolonjari, F., Wofsy, S. C., Pittman, J., and Daube, B. C.: Comparison of upper tropospheric carbon monoxide from MOPITT, ACE-FTS,

and HIPPO-QCLS, Journal of Geophysical Research-Atmospheres, 119, 14 144–14 164, https://doi.org/10.1002/2014JD022397, 2014.

Martínez-Alonso, S., Deeter, M., Worden, H., Borsdorff, T., Aben, I., Commane, R., Daube, B., Francis, G., George, M., Landgraf, J., Mao, D., McKain, K., and Wofsy, S.: 1.5 years of TROPOMI CO measurements: comparisons to MOPITT and ATom, Atmospheric Measurement Techniques, 13, 4841–4864, https://doi.org/10.5194/amt-13-4841-2020, 2020.

l. 29/38: at first mention of MOPITT and TROPOMII I suggest to include a reference to an instrumental description paper.l. 48: this is the first mentioning of AirCore. I suggest including a ref. here.

We have added these references for MOPITT (l. 29):

Drummond, J. and Mand, G.: The measurements of pollution in the troposphere (MOPITT) instrument: Overall performance and calibration requirements, Journal of Atmospheric and Oceanic Technology, 13, 314–320, https://doi.org/10.1175/1520-4300426(1996)013<0314:TMOPIT>2.0.CO;2, 1996.

Drummond, J. R., Zou, J., Nichitiu, F., Kar, J., Deschambaut, R., and Hackett, J.: A review of 9-year performance and operation of the MOPITT instrument, Advances in Space Research, 45, 760–774, https://doi.org/10.1016/j.asr.2009.11.019, 2010.

For TROPOMI (l. 38):

Veefkind, J. P., Aben, I., McMullan, K., Forster, H., de Vries, J., Otter, G., Claas, J., Eskes, H. J., de Haan, J. F., Kleipool, Q., van Weele, M., Hasekamp, O., Hoogeveen, R., Landgraf, J., Snel, R., Tol, P., Ingmann, P., Voors, R., Kruizinga, B., Vink, R., Visser, H., and Levelt, P. F.: TROPOMI on the ESA Sentinel-5 Precursor: A GMES mission for global observations of the atmospheric composition for climate, air quality and ozone layer applications, REMOTE SENSING OF ENVIRONMENT, 120, 70–83, https://doi.org/10.1016/j.rse.2011.09.027, 2012.

For AirCore (l. 48):

Karion, A., Sweeney, C., Tans, P., and Newberger, T.: AirCore: An Innovative Atmospheric Sampling System, Journal OF Atmospheric and Oceanic Technology, 27, 1839–1853, https://doi.org/10.1175/2010JTECHA1448.1, 2010.

Tans, P. P.: System and method for providing vertical profile measurements of atmospheric gases, US Patent Office, 2009.

l. 62: this dimension is specific for the NOAA AirCore and not a general AirCore feature. Other AirCores have different dimensions.

Thank you for pointing this out. We have reworded the text as follows: "The AirCore (Tans, 2009; Karion et al., 2010; Tans, 2021) is an innovative atmospheric sampling system comprised of a long tubing coil that is used to passively sample the atmosphere on high-altitude balloons."

l. 70: the altitude uncertainty is not only determined by diffusion. The flow into the tube and the time to establish a pressure equilibrium in the tube during fast descent is an important factor (see e.g. Wagenhaeuser et al., 2021).

We agree, thank you for pointing that out. We have reworded the text as follows: "While not empirically quantified in NOAA AirCore CO profiles presented here, others have quantified the uncertainty in AirCore altitude registration due to incorrect assumptions in AirCore tubing pressure equilibrium during balloon descent (Wagenhaeuser et al., 2021). NOAA profiles attempt to correct for pressure disequilibrium in the AirCore tubing and its effect on the total mass of air entering the AirCore at each altitude through comparisons of modeled pressure equilibrium and that measured *in situ* between ambient air and the closed end of the AirCore. This uncertainty is largest at lower atmospheric pressures (i.e., at more than ~20 km above mean sea level) than at higher ones. We hypothesize that this potential uncertainty component is likely to be of smaller magnitude than that calculated for CO

diffusion at altitudes up to 15-20 km (above which a higher uncertainty is likely, but these portions of the CO profiles are discarded as described below). Therefore, we believe this potential uncertainty component would have a negligible influence on the results presented here."

l. 92: again, assumptions on pressure equilibrium during sampling or modelling of the flow into AirCore play a role.
Thank you, we have reworded the text as follows: "Comparisons of stratospheric AirCore CO profiles have sometimes shown differences up to ~15 ppb, which could be a result of AirCore tubing surface interactions, uncertainties in the altitude registration of the AirCore CO profile, or diffusion effects."

l. 100: latter (not later)
Done, thank you for catching the typo.

l. 149: what is meant by the true value? Also AirCore has an uncertainty. I suggest to be more careful with the use of the term "true". This applies also to other places in the manuscript where "true" is used.
Thank you for pointing this out. We have removed/modified the word "true" when applied to AirCore profiles. We had explained in l. 321 and 322 that true AirCore profiles are unsmoothed AirCore profiles. However, we agree that the use of the term "true" in this case is somehow lax. The following occurrences of the word "true" have been modified as follows:
l. 20 from "AirCore true profiles" to "AirCore measured profiles"
l. 152 from "produce a 'smoothed' version of the true measurement" to "produce a 'smoothed' version of the *in situ* measurement"
l. 262 from "with respect to both true and smoothed AirCore" to "with respect to both unsmoothed and smoothed AirCore"
l. 321 from "biases between TROPOMI and true (i.e., unsmoothed) AirCore" to "biases between TROPOMI and unsmoothed AirCore"
l. 324 from "closer in value to true AirCore" to "closer in value to unsmoothed AirCore"
l. 265 from "for true and smoothed AirCore values" to "for unsmoothed and smoothed AirCore values"
Fig. 5 caption from "show true AirCore data" to "show unsmoothed AirCore data"

In other cases, however, the term "true" does apply and, thus, we have kept it (i.e., in Eq. 1 and its explanation, Eq. 2, Eq. 4, l. 271, l. 328, and l. 329). In these instances the text refers to the actual atmospheric composition at the time and location of the satellite observation, approximated with *in situ* measurements in practice. For clarity, we have added a few words to l. 150: "Xtrue is the true profile value (i.e., the actual atmospheric composition at the time and location of the remote observation, approximated in practice with *in situ* measurements)"

l. 175: please explain what null-space error is in a way that is understandable for a non-expert reader.
Thank you for making this point. We realize that such explanation was only provided further down in the manuscript, in l. 270: "The TROPOMI null-space error ($e_{null}$) is indicative of differences between the shape of TROPOMI CO reference profiles and that of true CO profiles which may result in differences between the true and TROPOMI-retrieved total CO column values." For clarity, we have moved that text to l. 178. To avoid repetitions, we have also reworded the text in l. 270 as follows: "The TROPOMI null-space error ($e_{null}$) quantifies the difference between the shapes of TROPOMI CO reference profiles and true CO profiles, which may result in differences between true and retrieved total CO column values."

l. 192: The value is certainly not 0.0 x 10^7. It is below a certain value. But not 0.

Reworded to "MOPITT TIR total column bias is below 0.1 x 10$^{17}$ molec cm$^{-2}$". For consistency, the corresponding entry in Table 1 has been modified similarly.

l. 227-234: I'm not usre that this part is really needed. This section should focus on the difference between using truncated and non-truncated AirCore profiles (i.e. what is discussed in lines 235-242).
We think that these lines are needed because they describe quantitatively MOPITT validation results when using "short" aircraft profiles (the truncated AirCore profiles, in this case) and demonstrate that the resulting biases are below MOPITT's target accuracy. From this baseline we move on, in the next paragraph, to describe and contrast MOPITT validation results when using the "full length" AirCore profiles.

Additionally, l. 227-234 indicate that biases are not the same for all three MOPITT variants (TIR, NIR, and multispectral). Some colleagues have recently pointed out to us that an explanation for these differences was missing in the manuscript. Thus, we have added the following text (l. 234): "In general, MOPITT multispectral products exhibit more extreme retrieval errors compared to TIR and NIR retrievals because the effects of potential radiance biases between measured and calculated radiances are amplified in the multispectral version of the retrieval algorithm. This amplification is done intentionally to boost the influence of the NIR radiances on the retrieval. In addition, multispectral retrievals are generally less stable than TIR and NIR retrievals because there is a greater chance that the radiances used in the retrieval will not be internally consistent (Deeter et al., 2012)"

l. 251: 2.16% of what?
According to Eq. 4, e_null=(I-A_c) *X_true. e_null in %=(100*e_null)/X_true.
For clarity, we have added to l. 251 a few words: "(2.16 % with respect to the *in situ* measurements)".